# Shape and Illumination from Shading using the Generic Viewpoint Assumption

**Daniel Zoran** [*]
CSAIL, MIT
danielz@mit.edu

**Dilip Krishnan** [*]
CSAIL, MIT
dilipkay@mit.edu

**Jose Bento**
Boston College
jose.bento@bc.edu

**William T. Freeman**
CSAIL, MIT
billf@mit.edu

## Abstract

The Generic Viewpoint Assumption (GVA) states that the position of the viewer or the light in a scene is not special. Thus, any estimated parameters from an observation should be stable under small perturbations such as object, viewpoint or light positions. The GVA has been analyzed and quantified in previous works, but has not been put to practical use in actual vision tasks. In this paper, we show how to utilize the GVA to estimate shape and illumination from a single shading image, without the use of other priors. We propose a novel linearized Spherical Harmonics (SH) shading model which enables us to obtain a computationally efficient form of the GVA term. Together with a data term, we build a model whose unknowns are shape and SH illumination. The model parameters are estimated using the Alternating Direction Method of Multipliers embedded in a multi-scale estimation framework. In this prior-free framework, we obtain competitive shape and illumination estimation results under a variety of models and lighting conditions, requiring fewer assumptions than competing methods.

## 1   Introduction

The generic viewpoint assumption (GVA) [5, 9, 21, 22] postulates that what we see in the world is not seen from a special viewpoint, or lighting condition. Figure 1 demonstrates this idea with the famous Necker cube example[1]. A three dimensional cube may be observed with two vertices or edges perfectly aligned, giving rise to a two dimensional interpretation. Another possibility is a view that exposes only one of the faces of the cube, giving rise to a square. However, these 2D views are unstable to slight perturbations in viewing position. Other examples in [9] and [22] show situations where views are unstable to lighting rotations.

While there has been interest in the GVA in the psychophysics community [22, 12], to the best of our knowledge, this principle seems to have been largely ignored in the computer vision community. One notable exception is the paper by Freeman [9] which gives a detailed analytical account on how to incorporate the GVA in a Bayesian framework. In that paper, it is shown that using the GVA modifies the probability space of different explanations to a scene, preferring perceptually valid and stable solutions to contrived and unstable ones, even though all of these fully explain the observed image. No algorithm incorporating the GVA, beyond exhaustive search, was proposed.

---

[*]Equal contribution
[1]Taken from http://www.cogsci.uci.edu/~ddhoff/three-cubes.gif

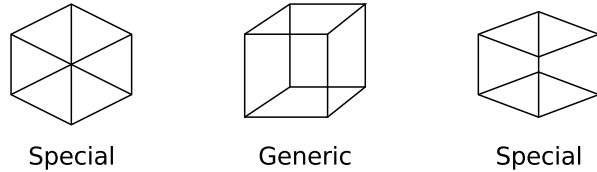

Figure 1: Illustration of the GVA principle using the Necker cube example. The cube in the middle can be viewed in multiple ways. However, the views on the left and right require a very specific viewing angle. Slight rotations of the viewer around the exact viewing positions would dramatically change the observed image. Thus, these views are unstable to perturbations. The middle view, on the contrary, is stable to viewer rotations.

Shape from shading is a basic low-level vision task. Given an input shading image - an image of a constant albedo object depicting only changes in illumination - we wish to infer the shape of the objects in the image. In other words, we wish to recover the relative depth $Z_i$ at each pixel $i$ in the image. Given values of $Z$, local surface orientations are given by the gradients $\nabla_x Z$ and $\nabla_y Z$ along the coordinate axes. A key component in estimating the shape is the illumination $\mathbf{L}$. The parameters of $\mathbf{L}$ may be given with the image, or may need to be estimated from the image along with the shape. The latter is a much harder problem due to the ambiguous nature of the problem, as many different surface orientations and light combinations may explain the same image. While the notion of a shading image may seem unnatural, extracting them from natural images has been an active field of research. There are effective ways of decomposing images into shading and albedo images (so called "intrinsic images" [20, 10, 1, 29]), and the output of those may be used as input to shape from shading algorithms.

In this paper we show how to effectively utilize the GVA for shape and illumination estimation from a single shading image. The only terms in our optimization are the data term which explains the observation and the GVA term. We propose a novel shading model which is a linearization of the spherical harmonics (SH) shading model [25]. The SH model has been gaining popularity in the vision and graphics communities in recent years [26, 17]) as it is more expressive than the popular single source Lambertian model. Linearizing this model allows us, as we show below, to get simple expressions for our image and GVA terms, enabling us to use them effectively in an optimization framework. Given a shading image with an unknown light source, our optimization procedure solves for the depth and illumination in the scene. We optimize using Alternating Direction Method of Multipliers (ADMM) [4, 6]. We show that this method is competitive with current shape and illumination from shading algorithms, without the use of other priors over illumination or geometry.

## 2 Related Work

Classical works on shape from shading include [13, 14, 15, 8, 23] and newer works include [3, 2, 19, 30]. It is out of scope of this paper to give a full survey of this well studied field, and we refer the reader to [31] and [28] for good reviews. A large part of the research has been focused on estimating the shape under known illumination conditions. While still a hard problem, it is more constrained than estimating both the illumination and the shape.

In impressive recent work, Barron and Malik [3] propose a method for estimating not just the illumination and shape, but also the albedo of a given masked object from a single image. By using a number of novel (and carefully balanced) priors over shape (such as smoothness and contour information), albedo and illumination, it is shown that reasonable estimates of shape and illumination may be extracted. These priors and the data term are combined in a novel multi-scale framework which weights coarser scale (lower frequency) estimates of shape more than finer scale estimates. Furthermore, Barron and Malik use a spherical harmonics lighting model to provide for richer recovery of real world scenes and diffuse outdoor lighting conditions. Another contribution of their work has been the observation that joint inference of multiple parameters may prove to be more robust (although this is hard to prove rigorously). The expansion to the original MIT dataset [11] provided in [3] is also a useful contribution.

Another recent notable example is that of Xiong et al. [30]. In this thorough work, the distribution of possible shape/illumination combinations in a small image patch is derived, assuming a quadratic depth model. It is shown that local patches may be quite informative, and that are only a few possible explanations of light/shape pairs for each patch. A framework for estimating full model geometry with *known* lighting conditions is also proposed.

## 3 Using the Generic View Assumption for Shape from Shading

In [9], Freeman gave an analytical framework to use the GVA. However, the computational examples in the paper were restricted to linear shape from shading models. No inference algorithm was presented; instead the emphasis was on analyzing how the GVA term modifies the posterior distribution of candidate shape and illumination estimates. The key idea in [9] is to marginalize the posterior distribution over a set of "nuisance" parameters - these correspond to object or illumination perturbations. This integration step corresponds to finding a solution that is stable to these perturbations.

### 3.1 A Short Introduction to the GVA

Here we give a short summary of the derivations in [9], which we use in our model. We start with a generative model $\mathbf{f}$ for images, which depends on scene parameters $\mathbf{Q}$ and a set of generic parameters $\mathbf{w}$. The generative model we use is explained in Section 4. $\mathbf{w}$ are the parameters which will eventually be marginalized. In our shape and illumination from shading case, $\mathbf{f}$ corresponds to our shading model in Eq. 14 (defined below). $\mathbf{Q}$ includes both surface depth at each point $\mathbf{Z}$ and the light coefficients vector $\mathbf{L}$. Finally, the generic variable $\mathbf{w}$ corresponds to different object rotation angles around different axes of rotations (though there could be other generic variables, we only use this one). Assuming measurement noise $\eta$ the result of the generative process would be:

$$\mathbf{I} = \mathbf{f}(\mathbf{Q}, \mathbf{w}) + \eta \tag{1}$$

Now, given an image $\mathbf{I}$ we wish to infer scene parameters $\mathbf{Q}$ by marginalizing out the generic variables $\mathbf{w}$. Using Bayes' theorem, this results in the following probability function:

$$P(\mathbf{Q}|\mathbf{I}) = \frac{P(\mathbf{Q})}{P(\mathbf{I})} \int_{\mathbf{w}} P(\mathbf{w}) P(\mathbf{I}|\mathbf{Q}, \mathbf{w}) d\mathbf{w} \tag{2}$$

Assuming a low Gaussian noise model for $\eta$, the above integral can be approximated with a Laplace approximation, which involves expanding $\mathbf{f}$ using a Taylor expansion around $\mathbf{w}_0$. We get the following expression, aptly named in [9] as the "scene probability equation":

$$P(\mathbf{Q}|\mathbf{I}) = \underbrace{C}_{\text{constant}} \underbrace{\exp\left(-\frac{\|\mathbf{I} - \mathbf{f}(\mathbf{Q}, \mathbf{w}_0)\|^2}{2\sigma^2}\right)}_{\text{fidelity}} \underbrace{P(\mathbf{Q})P(\mathbf{w}_0)}_{\text{prior}} \underbrace{\frac{1}{\sqrt{\det \mathbf{A}}}}_{\text{genericity}} \tag{3}$$

where $\mathbf{A}$ is a matrix whose $i, j$-th entry is:

$$A_{i,j} = \frac{\partial \mathbf{f}(\mathbf{Q}, \mathbf{w})}{\partial w_i}^T \frac{\partial \mathbf{f}(\mathbf{Q}, \mathbf{w})}{\partial w_j} \tag{4}$$

and the derivatives are estimated at $\mathbf{w}_0$. $A$ is often called the Fisher information matrix.

Eq. 3 has three terms: the fidelity term (sometimes called the likelihood term, data term or image term) tells us how close we are to the observed image. The prior tells us how likely are our current parameter estimates. The last term, genericity, tells us how much our observed image would change under perturbations of the different generic variables. This term is the one which penalizes for unstable results w.r.t to the generic variables. From the form of $A$, it is clear why the genericity term helps; the determinant of $A$ is large when the rendered image $f$ changes rapidly with respect to $w$. This makes the genericity term small and the corresponding hypothesis $\mathbf{Q}$ less probable.

## 3.2 Using the GVA for Shape and Illumination Estimation

We now show how to derive the GVA term for general object rotations by using the result in [9] and applying it to our linearized shading model. Due to lack of space, we provide the main results here; please refer to the supplementary material for full details. Given an axis of rotation parametrized by angles $\theta$ and $\gamma$, the derivative of $\mathbf{f}$ w.r.t to a rotation $\phi$ about the axis is:

$$\frac{\partial \mathbf{f}}{\partial \phi} = a\mathbf{R}^x + b\mathbf{R}^y + c\mathbf{R}^z \tag{5}$$

$$a = \cos(\theta)\sin(\gamma), \quad b = \sin(\theta)\sin(\gamma), \quad c = \cos(\gamma) \tag{6}$$

where $\mathbf{R}^x$, $\mathbf{R}^y$ and $\mathbf{R}^z$ are three derivative images for rotations around the canonical axes for which the $i$-th pixel is:

$$\mathbf{R}_i^x = I_i^x Z_i + \alpha_i \beta_i k_i^x + (1 + \beta_i^2)k_i^y \tag{7}$$

$$\mathbf{R}_i^y = -I_i^y Z_i - \alpha_i \beta_i k_i^y - (1 + \alpha_i^2)k_i^x \tag{8}$$

$$\mathbf{R}_i^z = I_i^x Y_i - I_i^y X_i + \alpha_i k_i^y - \beta_i k_i^x \tag{9}$$

We use these images to derive the GVA term for rotations around different axes, resulting in:

$$\text{GVA}(\mathbf{Z}, \mathbf{L}) = \sum_{\theta \in \Theta} \sum_{\gamma \in \Gamma} \frac{1}{\sqrt{2\pi\sigma^2 \|\frac{\partial \mathbf{f}}{\partial \phi}\|^2}} \tag{10}$$

where $\Theta$ and $\Gamma$ are discrete sets of angles in $[0, \pi)$ and $[0, 2\pi)$ respectively. Looking at the term in Eqs. 5–10 we see that had we used the full, non-linearized, shading model in Eq. 11 it would result in a very complex expression, especially considering that $\alpha = \nabla_x \mathbf{Z}$ and $\beta = \nabla_y \mathbf{Z}$ are functions of the depth $\mathbf{Z}$. Even after linearization, this expression may seem a bit daunting, but we show in Section 5 how we can significantly simplify the optimization of this function.

## 4 Linearized Spherical Harmonics Shading Model

The Spherical Harmonics (SH) lighting[2] model allows for a rich yet concise description of a lighting environment [25]. By keeping just a few of the leading SH coefficients when describing the illumination, it allows an accurate description for low frequency changes of lighting as a function of direction, without needing to explicitly model the lighting environment in whole. This model has been used successfully in the graphics and the vision communities. The popular setting for SH lighting is to keep the first three orders of the SH functions, resulting in nine coefficients which we will denote by the vector $\mathbf{L}$. Let $Z$ be a depth map, with the depth at pixel $i$ given by $Z_i$. The surface slopes at pixel $i$ are defined as $\alpha_i = (\nabla_x Z)_i$ and $\beta_i = (\nabla_y Z)_i$ respectively. Given $\mathbf{L}$ and $Z$, the log shading at pixel $i$ for a diffuse, Lambertian surface under the SH model is given by:

$$\log S_i = \mathbf{n}_i^T \mathbf{M} \mathbf{n}_i \tag{11}$$

where $\mathbf{n}_i$:

$$\mathbf{n}_i = \begin{bmatrix} \frac{\alpha_i}{\sqrt{1+\alpha_i^2+\beta_i^2}} & \frac{\beta_i}{\sqrt{1+\alpha_i^2+\beta_i^2}} & \frac{1}{\sqrt{1+\alpha_i^2+\beta_i^2}} & 1 \end{bmatrix}^T \tag{12}$$

and:

$$\mathbf{M} = \begin{bmatrix} c_1 L_9 & c_1 L_5 & c_1 L_8 & c_2 L_4 \\ c_1 L_5 & -c_1 L_9 & c_1 L_6 & c_2 L_2 \\ c_1 L_8 & c_1 L_6 & c_3 L_7 & c_2 L_3 \\ c_2 L_4 & c_2 L_2 & c_2 L_3 & c_4 L_1 - c_5 L_7 \end{bmatrix} \tag{13}$$

$$c_1 = 0.429043, c_2 = 0.511664, c_3 = 0.743125, c_4 = 0.886227, c_5 = 0.247708$$

The formation model in Eq. 11 is non-linear and non-convex in the surface slopes $\alpha$ and $\beta$. In practice, this leads to optimization difficulties such as local minima, which have been noted by Barron and Malik in [3]. In order to overcome this, we linearize Eq. 11 around the local surface slope estimate $\alpha_i^0$ and $\beta_i^0$, such that:

$$\log S_i \approx k^c(\alpha_i^0, \beta_i^0, \mathbf{L}) + k^x(\alpha_i^0, \beta_i^0, \mathbf{L})\alpha_i + k^y(\alpha_i^0, \beta_i^0, \mathbf{L})\beta_i \tag{14}$$

where the local surface slopes are estimated in a local patch around each pixel in our current estimated surface. The derivation of the linearization is given in the supplementary material. For the sake of brevity, we will omit the dependence on the $\alpha_i^0, \beta_i^0$ and $\mathbf{L}$ terms, and denote the coefficients at each location as $k_i^c, k_i^x$ and $k_i^y$ respectively for the remainder of the paper.

A natural question is the accuracy of the linearized model Eq. 14. The linearization is accurate in most situations where the depth $Z$ changes gradually, such that the change in slope is linear or small in magnitude. In [30], locally quadratic shapes are assumed; this leads to linear changes in slopes, and in such situations, the linearization is highly accurate. We tested the accuracy of the linearization by computing the difference between the estimates in Eq. 14 and Eq. 11, over ground truth shape and illumination estimates. We found it to be highly accurate for the models in our experiments. The linearization in Eq. 14 leads to a quadratic formation model for the image term (described in Section 5.2.1), leading to more efficient updates for $\alpha$ and $\beta$. Furthermore, this allows us to effectively incorporate the GVA even with the spherical harmonics framework.

# 5   Optimization using the Alternating Direction Method of Multipliers

## 5.1   The Cost Function

Following Eq. 3, we can now derive the cost function we will optimize w.r.t the scene parameters $\mathbf{Z}$ and $\mathbf{L}$. To derive a MAP estimate, we take the negative $\log$ of Eq. 3 and use constant priors over both the scene parameters and the generic variables; thus we have a prior-free cost function. This results in the following cost:

$$g(\mathbf{Z}, \mathbf{L}) = \lambda_{\text{img}} \|\mathbf{I} - \log S(\mathbf{Z}, \mathbf{L})\|^2 - \lambda_{\text{GVA}} \log \text{GVA}(\mathbf{Z}, \mathbf{L}) \tag{15}$$

where $\mathbf{f}(\mathbf{Z}, \mathbf{L}) = \log S(\mathbf{Z}, \mathbf{L})$ is our linearized shading model Eq. 14 and the GVA term is defined in Eq. 10. $\lambda_{\text{img}}$ and $\lambda_{\text{GVA}}$ are hyper-parameters which we set to 2 and 1 respectively for all experiments. Because of the dependence of $\alpha$ and $\beta$ on $\mathbf{Z}$ directly optimizing for this cost function is hard, as it results in a large, non-linear differential system for $\mathbf{Z}$. In order to make this more tractable, we introduce $\tilde{\alpha}$ and $\tilde{\beta}$, the surface spatial derivatives, as auxiliary variables, and solve for the following cost function which constrains the resulting surface to be integrable:

$$\tilde{g}(\mathbf{Z}, \tilde{\alpha}, \tilde{\beta}, \mathbf{L}|\mathbf{I}) = \lambda_{\text{img}} \|\mathbf{I} - \log S(\tilde{\alpha}, \tilde{\beta}, \mathbf{L})\|^2 - \lambda_{\text{GVA}} \log \text{GVA}(\mathbf{Z}, \tilde{\alpha}, \tilde{\beta}, \mathbf{L}) \tag{16}$$

$$\text{s.t} \quad \tilde{\alpha} = \nabla_x \mathbf{Z}, \quad \tilde{\beta} = \nabla_y \mathbf{Z}, \quad \nabla_y \nabla_x \mathbf{Z} = \nabla_x \nabla_y \mathbf{Z}$$

ADMM allows us to subdivide the cost into relatively simple subproblems, solve each one independently and then aggregate the results. We briefly review the message passing variant of ADMM [7] in the supplementary material.

## 5.2   Subproblems

### 5.2.1   Image Term

This subproblem ties our solution to the input log shading image. The participating variables are the slopes $\tilde{\alpha}$ and $\tilde{\beta}$ and illumination $\mathbf{L}$. We minimize the following cost:

$$\underset{\tilde{\alpha}, \tilde{\beta}, \mathbf{L}}{\arg\min} \; \lambda_{\text{img}} \sum_i \left( I_i - k_i^c - k_i^x \tilde{\alpha}_i - k_i^y \tilde{\beta}_i \right)^2 + \frac{\rho}{2} \|\tilde{\alpha} - n_{\tilde{\alpha}}\|^2 + \frac{\rho}{2} \|\tilde{\beta} - n_{\tilde{\beta}}\|^2 + \frac{\rho}{2} \|\mathbf{L} - n_{\mathbf{L}}\|^2 \tag{17}$$

where $n_{\tilde{\alpha}}, n_{\tilde{\beta}}$ and $n_{\mathbf{L}}$ are the incoming messages for the corresponding variables as described above. We solve this subproblem iteratively: for $\tilde{\alpha}$ and $\tilde{\beta}$ we keep $\mathbf{L}$ constant (and as a result the $k$-s are constant). A closed form solution exists since this is just a quadratic due to our relinearization model. In order to solve for $\mathbf{L}$ we do a few (5 to 10) steps of L-BFGS [27].

### 5.2.2   GVA Term

The participating variables here are the depth values $\mathbf{Z}$, the slopes $\tilde{\alpha}$ and $\tilde{\beta}$ and the light $\mathbf{L}$. We look for the parameters which minimize:

$$\underset{\mathbf{Z}, \tilde{\alpha}, \tilde{\beta}, \mathbf{L}}{\arg\min} \; -\frac{\lambda_{\text{GVA}}}{2} \log \text{GVA}(\mathbf{Z}, \tilde{\alpha}, \tilde{\beta}, \mathbf{L}) + \frac{\rho}{2} \|\tilde{\alpha} - n_{\tilde{\alpha}}\|^2 + \frac{\rho}{2} \|\tilde{\beta} - n_{\tilde{\beta}}\|^2 + \frac{\rho}{2} \|\mathbf{L} - n_{\mathbf{L}}\|^2 \tag{18}$$

Here, though the expression for the GVA (Eq. 10) term is greatly simplified due to the shading model linearization, we have to resort to numerical optimization. We solve for the parameters using a few steps of L-BFGS [27].

### 5.2.3 Depth Integrability Constraint

Shading only depends on local slope (regardless of the choice of shading model, as long as there are no shadows in the scene), hence the image term only gives us information about surface slopes. Using this information we need to find an integrable surface $\mathbf{Z}$ [8]. Finding integrable surfaces from local slope measurements has been a long standing research question and there are several ways of doing this [8, 14, 18]. By finding such as a surface we will satisfy both constraints in Eq. 16 automatically. Enforcing integrability through message passing was performed in [24], where it was shown to be helpful in recovering smooth surfaces. In that work, belief propagation based message-passing was used. The cost for this subproblem is:

$$\underset{\mathbf{Z},\tilde{\alpha},\tilde{\beta}}{\arg\min} \frac{\rho}{2}\|\mathbf{Z} - n_{\mathbf{Z}}\|^2 + \frac{\rho}{2}\|\tilde{\alpha} - n_{\tilde{\alpha}}\|^2 + \frac{\rho}{2}\|\tilde{\beta} - n_{\tilde{\beta}}\|^2 \tag{19}$$

$$\text{s.t} \quad \tilde{\alpha} = \nabla_x\mathbf{Z}, \quad \tilde{\beta} = \nabla_y\mathbf{Z}, \quad \nabla_y\nabla_x\mathbf{Z} = \nabla_x\nabla_y\mathbf{Z}$$

We solve for the surface $\mathbf{Z}$ given the messages for the slopes $n_{\tilde{\alpha}}$ and $n_{\tilde{\beta}}$ by solving a least squares system to get the integrable surface. Then, the solution for $\tilde{\alpha}$ and $\tilde{\beta}$ is just the spatial derivative of the resulting surface, satisfying all the constraints and minimizing the cost simultaneously.

### 5.3 Relinearization

After each ADMM iteration, we perform re-linearization of the $k_c$, $k_x$ and $k_y$ coefficients. We take the current estimates for $\mathbf{Z}$ and $\mathbf{L}$ and use them as input to our linearization procedure (see the supplementary material for details). These coefficients are then used for the next ADMM iteration. and this process is repeated.

## 6 Experiments and Results

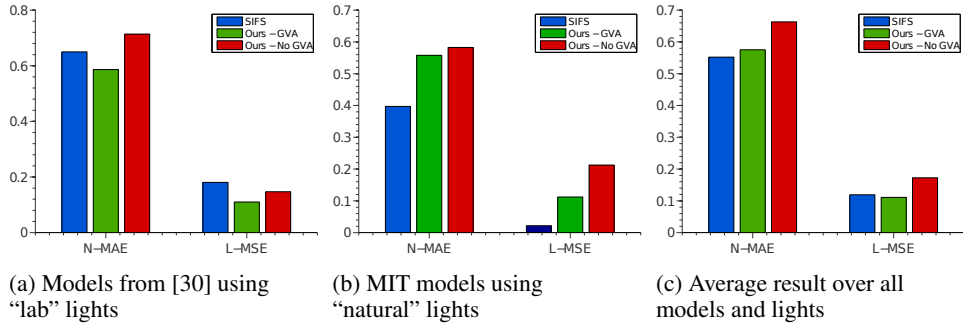

(a) Models from [30] using "lab" lights

(b) MIT models using "natural" lights

(c) Average result over all models and lights

Figure 2: Summary of results: Our performance is quite similar to that of SIFS [3] although we do not use contour normals, nor any shape or illumination priors unlike [3]. We outperform SIFS on models from [30], while SIFS performs well on the MIT models. On average, we are comparable to SIFS in N-MAE and sightly better at light estimation.

We use the GVA algorithm to estimate shape and illumination from synthetic, grayscale shading images, rendered using 18 different models from the MIT/Berkeley intrinsic images dataset [3] and 7 models from the Harvard dataset in [30]. Each of these models is rendered using several different light sources: the MIT models are lit with a "natural" light dataset which comes with each model, and we use 2 lights from the "lab" dataset in order to light the models from [30], resulting in 32 different images. We use the provided mask just in the image term, where we solve only for pixels within the mask. We do not use any other contour information as in [3]. Models were downscaled to a quarter of their original size. Running times for our algorithm are roughly 7 minutes per image

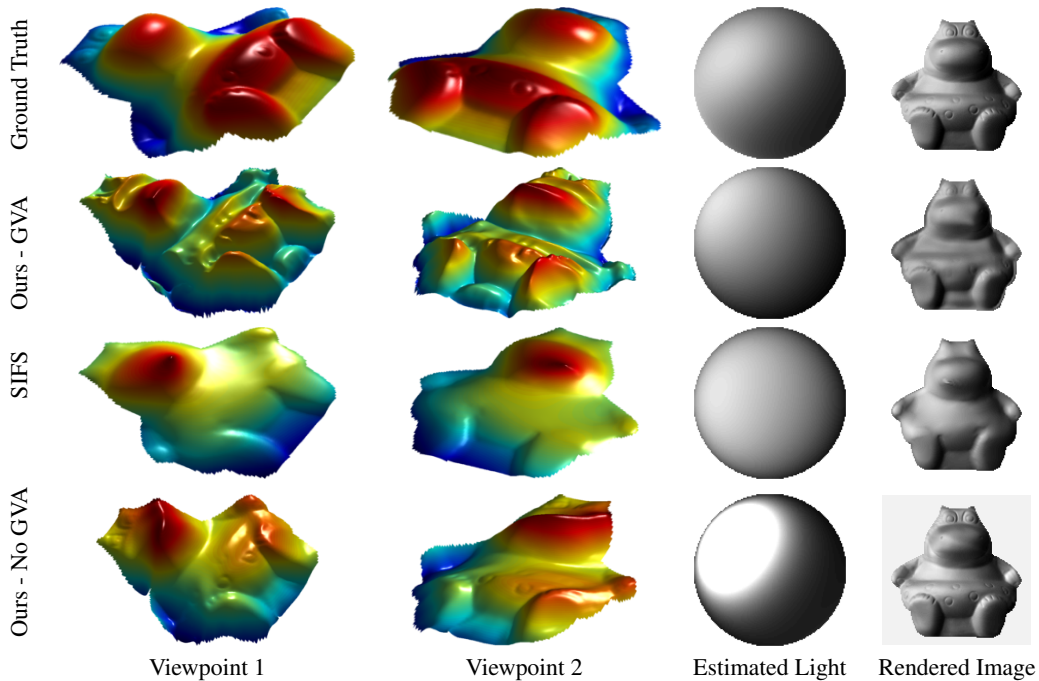

Figure 3: Example of our results - note that the vertical scale of the mesh plots is different between the plots and have been rescaled for display (specifically, the SIFS result are 4 times deeper). Our method preserves features such as the legs and belly while SIFS smoothes them out. The GVA light estimate is also quite reasonable. Unlike SIFS, no contour normals, nor tuned shape or lighting priors are needed for GVA.

with the GVA term and about 1 minute without the GVA term. This is with unoptimized `MATLAB` code. We compare to the SIFS algorithm of [3] which is a subset of their algorithm that does not estimate albedo. We use their publicly released code.

We initialize with an all zeros depth (corresponding to a flat surface) and the light is initialized to the mean light from the "natural" dataset in [3]. We perform the estimation in multiple scales using V-sweeps - solving at a coarse scale, upscaling, solving at a finer scale then downsampling the result, repeating the process 3 times. The same parameter settings were used in all cases[3].

We use the same error measures as in [3]. The error for the normals is measured using Median Angular Error (MAE) in radians. For the light, we take the resulting light coefficients and render a sphere lit by this light. We look for a DC shift which minimizes the distance between this image and the rendered ground truth light and shift the two images. Then the final error for the light is the $L_2$ distance of the two images, normalized by the number of pixels. The error measure for depth $\mathbf{Z}$ used in [3] is quite sensitive to the absolute scaling of the results. We have decided to omit it from the main paper (even though our performance under this measure is much better than [3]).

A summary of the results can be seen in Figure 2. The GVA term helps significantly in estimation results. This is especially true for light estimation. On average, our performance is similar to that of [3]. Our light estimation results are somewhat better, while our geometry estimation results are slightly poorer. It seems that [3] is somewhat overfit to the models in the MIT dataset. When tested on the models from [30], it gets poorer results.

Figure 3 shows an example of the results we get, compared to that of SIFS [3], our algorithm with no GVA term, and the ground truth. As can be seen, the light we estimate is quite close to the ground truth. The geometry we estimate certainly captures the main structures of the ground truth. Even though we use no smoothness prior, the resulting mesh is acceptable - though a smoothness prior, such as the one used [3] would help significantly. The result by [3] misses a lot of the large

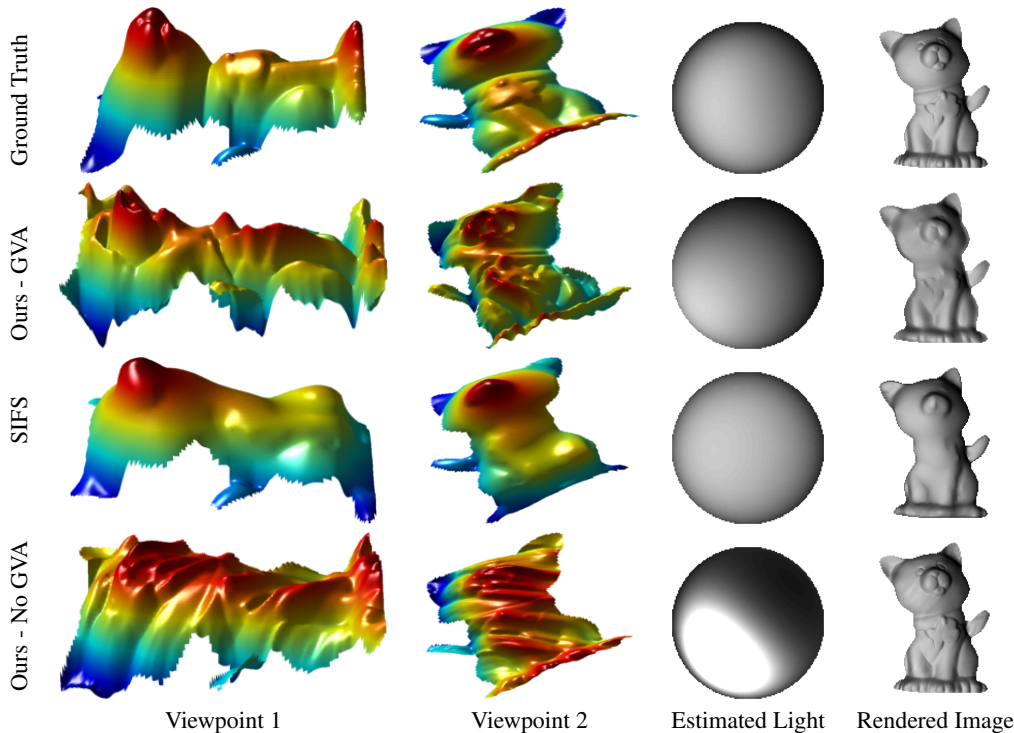

Figure 4: Another example. Note how we manage to recover some of the dominant structure like the neck and feet, while SIFS mostly smooths features (albeit resulting in a more pleasing surface).

scale structures of such as the hippo's belly and feet, but it is certainly smooth and aesthetic. It is seen that without the GVA term, the resulting light is highly directed and the recovered shape has snake-like structures which precisely line up with the direction of the light. These are very specific local minima which satisfy the observed image well, in agreement with the results in [9]. Figure 4 shows some more results on a different model where the general story is similar.

## 7  Discussion

In this paper, we have presented a shape and illumination from shading algorithm which makes use of the Generic View Assumption. We have shown how to utilize the GVA within an optimization framework. We achieve competitive results on shape and illumination estimation without the use of shape or illumination priors. The central message of our work is that the GVA can be a powerful regularizing term for the shape from shading problem. While priors for scene parameters can be very useful, balancing the effect of different priors can be hard and inferred results may be biased towards a wrong solution. One may ask: is the GVA just another prior? The GVA is a prior assumption, but a very reasonable one: it merely states that all viewpoints and lighting directions are equally likely. Nevertheless, there may exist multiple stable solutions and priors may be necessary to enable choosing between these solutions [16]. A classical example of this is the convex/concave ambiguity in shape and light.

Future directions for this work are applying the GVA to more vision tasks, utilizing better optimization techniques and investigating the coexistence of priors and GVA terms.

### Acknowledgments

This work was supported by NSF CISE/IIS award 1212928 and by the Qatar Computing Research Institute. We would like to thank Jonathan Yedidia for fruitful discussions.

## Footnotes

[2]We will use the terms lighting and shading interchangeably

[3]We will make our code publicly available at `http://dilipkay.wordpress.com/sfs/`

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
