[Supplementary Material]

# Shape and Illumination From Shading Using the Generic Viewpoint Assumption - Supplementary Material

**Daniel Zoran** *
CSAIL, MIT
danielz@mit.edu

**Dilip Krishnan** *
CSAIL, MIT
dilipkay@mit.edu

**Jose Bento**
Boston College
jose.bento@bc.edu

**William T. Freeman**
CSAIL, MIT
billf@mit.edu

In this supplementary material, we provide various technical details for our paper. The latest copy of the paper and this supplementary material will be available on the project webpage http:// dilipkay.wordpress.com/sfs/.

## 1 Depth Error Measure

In Table 1, we show the errors between ground truth depth and our depth measurements using the Z-MAE error measure introduced in [1]. It can be seen that our algorithms have consistently lower error. However, this error measure is highly dependent on the absolute range of values in the recovered depth $Z$. The error measure is not consistent with perceptual assessment of recovered depth quality, and so we report it here only for completeness.

| Method | Natural Lights | Lab Lights | All Lights |
|--------|:--------------:|:----------:|:----------:|
| SIFS | 11.5 | 10.7 | 11.3 |
| Ours - GVA | 8.3 | 6.0 | 7.8 |
| Ours - no GVA | 8.4 | 6.2 | 7.9 |

Table 1: Median Error of recovered depth values (called Z-MAE in [1]).

## 2 Derivation of Linearization Formula

As shown in our paper, the effectiveness of using GVA in a computational framework relies heavily on the relinearization of the Spherical Harmonics lighting model. The original non-linear model that generates the log shading pixels is given by the following per-pixel equation (each pixel is independent of the others):

$$h(\alpha, \beta) = \mathbf{n}^T \mathbf{M} \mathbf{n} \tag{1}$$

where $\mathbf{n}$ is given by:

$$\mathbf{n} = \left[ \begin{array}{cccc} \frac{\alpha}{\sqrt{1+\alpha^2+\beta^2}} & \frac{\beta}{\sqrt{1+\alpha^2+\beta^2}} & \frac{1}{\sqrt{1+\alpha^2+\beta^2}} & 1 \end{array} \right] \tag{2}$$

---

and:

$$\mathbf{M} = \begin{bmatrix} c_1 L_9 & c_1 L_5 & c_1 L_8 & c_2 L_4 \\ c_1 L_5 & -c_1 L_9 & c_1 L_6 & c_2 L_2 \\ c_1 L_8 & c_1 L_6 & c_3 L_7 & c_2 L_3 \\ c_2 L_4 & c_2 L_2 & c_2 L_3 & c_4 L_1 - c_5 L_7 \end{bmatrix} \tag{3}$$

where $c_1 = 0.429043$, $c_2 = 0.511664$, $c_3 = 0.743125$, $c_4 = 0.886227$ and $c_5 = 0.247708$. Note that $\alpha$ and $\beta$ are scalars. Also for later use, let us define the vector $\mathbf{x}$ such that:

$$\mathbf{n} = \begin{bmatrix} \mathbf{x} & 1 \end{bmatrix} \tag{4}$$

This model is linear in $L$ (since $M$ is linear in $L$) and quadratic in $\mathbf{n}$. Given a current estimate $\alpha^0$ and $\beta^0$ for the surface slopes, the Taylor series expansion of $h$ around $(\alpha_0, \beta_0)$ is given by:

$$h(\alpha, \beta) = h(\alpha^0, \beta^0) + [\nabla_\alpha h \quad \nabla_\beta h]\Big|_{(\alpha^0, \beta^0)} [\alpha - \alpha^0 \quad \beta - \beta^0]^T + \text{higher order terms} \tag{5}$$

Here the gradient of $h$ is with respect to $\alpha$ and $\beta$. Now we derive these terms individually using the chain rule:

$$\nabla_\alpha h = \nabla_\mathbf{x} f \ \nabla_\alpha \mathbf{x}$$
$$\nabla_\beta h = \nabla_\mathbf{x} f \ \nabla_\beta \mathbf{x}$$

where $\mathbf{x}$ is defined in Eq. 4. Let $M_1$ be the top left $3 \times 3$ block of $M$ and $M_2$ be the first 3 columns of the last row of $M$ (note that $M$ is symmetric). Then we get:

$$\nabla_\mathbf{x} h = [2\mathbf{x} M_1 + 2 M_2] \tag{6}$$

$$\nabla_\alpha \mathbf{x} = \begin{bmatrix} \frac{1}{\sqrt{1+\alpha^2+\beta^2}} - \frac{\alpha^2}{\sqrt{(1+\alpha^2+\beta^2)^3}} & \frac{-\alpha\beta}{\sqrt{(1+\alpha^2+\beta^2)^3}} & \frac{-\alpha}{\sqrt{(1+\alpha^2+\beta^2)^3}} \end{bmatrix} \tag{7}$$

$$\nabla_\beta \mathbf{x} = \begin{bmatrix} \frac{-\alpha\beta}{\sqrt{(1+\alpha^2+\beta^2)^3}} & \frac{1}{\sqrt{1+\alpha^2+\beta^2}} - \frac{\beta^2}{(1+\alpha^2+\beta^2)^3} & \frac{-\beta}{\sqrt{(1+\alpha^2+\beta^2)^3}} \end{bmatrix} \tag{8}$$

This gives us the values for $\nabla_\alpha h$ and $\nabla_\beta h$, which can then be substituted in Eq. 5. Collecting all constants into $k_c$ and $\alpha$ and $\beta$ coefficients into $k_x$ and $k_y$ respectively gives the final linearized model:

$$h(\alpha, \beta) \approx k^c + k^x \alpha + k^y \beta \tag{9}$$

## 3  Introduction to ADMM

The Alternating Direction Method of Multipliers has been developed in various forms over the last few decades. Initially, it was introduced in the 1970s for the numerical solution of partial differential equations. ADMM is closely related to the ideas of dual decomposition and augmented Lagrangians. A thorough review of these ideas is given in [3]. Bento et al. [2] introduced a message-passing version of ADMM, which is equivalent to the classical ADMM method when only a single weight is used. However, by using different kinds of messages, they are able to significantly improve the convergence rate of ADMM for certain constrained problems.

We give here a brief description of the message passing ADMM. Further details can be found in the above references. A simple canonical problem we consider is the following:

$$\min f(x) + g(z)$$
$$\text{s.t.} \quad x = z \tag{10}$$

This problem arises in many contexts, for example when $f$ is a likelihood function and $g$ is a regularization term. Message-passing ADMM now considers the above problem to consist of 3 sub-problems, each of which can be solved in parallel. The three sub-problems involve $f(x)$, $g(z)$ and

the constraint $x = z$ respectively. Thus the variable $x$ is involved in 2 sub-problems, and the variable $z$ in two sub-problems. There are two conflicting requirements to ensure that we make progress: each sub-problem must make progress towards minimizing it's own cost function; and secondly, it must not provide a solution that is completely different from the other sub-problem that involves the same variables.

This tension is resolved in message-passing ADMM by the use of *regularized* sub-problems which ensure that the solution of the sub-problem does not move too far away from the current consensus solution for the set of variables involved in that sub-problem. The strength of regularization is controlled by a parameter $\rho$. The regularized sub-problems for the above canonical problem are given by:

$$\min f(x) + \frac{\rho}{2}(x - n_x)^2$$
$$\min g(z) + \frac{\rho}{2}(z - n_z)^2$$
$$\min \frac{\rho}{2}(x - n_x)^2 + \frac{\rho}{2}(z - n_z)^2 \text{ s.t. } x = z$$

The "messages" $n_x$ and $n_z$ encode the current consensus for the $x$ and $z$ variables respectively. Larger $\rho$ values ensure that the sub-problems do not move too far away from the consensus; however, this could slow down convergence. Smaller $\rho$ values allow each sub-problem to get to a better local minimum; however, this might cause significant oscillation in the consensus values. $\rho$ is a hyper-parameter who's value is problem-dependent.

Further details of the consensus mechanism and the structure of the messages $n_x$ and $n_z$ are given in [2]. The key benefit of ADMM is that individual sub-problems may be solved in parallel and admit fast solutions; for example if $f(x)$ is a quadratic and $g(z)$ is an $l_1$ regularization term. Secondly, the lifting of variables to a higher-dimensional space may enable escaping local minima in the case of non-convex problems.