[Reviews · NeurIPS 2014]

Submitted by Assigned_Reviewer_3

This paper presents a very novel and interesting algorithm for shape-from-shading in unknown illumination. Because SFS is inherently underconstrained even when illumination is known, any SFS algorithm must appeal to some prior assumptions --- usually in the form of smoothness constraints on surfaces. This paper, instead, revives the old notion of the generic viewpoint assumption: that the appearance of an object should be stable under small perturbations. Surprisingly, the authors show that this "prior" is roughly as effective as a more conventional state-of-the-art shape-from-shading algorithm. This is especially surprising given that the GVA is an old idea which has never seemed to be a useful insight for solving real problems. The success of this technique is also impressive because there is only one other SFS algorithm ([3]) that can cope with unknown illumination, despite how old and ubiquitous this problem is.

Clarity: The paper is well written and as clear as can be, given the challenging and dense nature of this material.

Originality: This paper is very original in both its high-level vision and its low-level technical details. As a vision paper, the overall approach of using the generic viewpoint assumption to drive a SFS algorithm is very novel. As an inference/optimization paper, it also appears to be very novel and interesting, though I am less well-qualified to comment on these aspects as a reviewer.

Significance: This paper will certainly affect the shape-from-shading community, as it demonstrates a very powerful prior that I imagine would be useful in other frameworks. I could imagine this paper also having a consider impact in the SFS / shape reconstruction literature by calling attention to the various optimization tricks used in this paper (linearized spherical harmonic illumination, ADMM for surface normal reconstruction). This paper might serve to highlight the fact that these more classic computer vision problems are fertile grounds for inference/optimization people to apply their tools.

My only complaints with this paper are:
- It only addresses a subset of the problem of [3], where albedo is assumed to be constant. This is reasonable given that this is the first attempt at this problem, but it is unfortunate that this paper only roughly matches the performance of [3], while solving a much easier version of the problem that [3] was designed for. I still think this paper should be accepted, if for no other reason than to allow others to build towards a more complete algorithm that tackles the greater problem.
- The authors are quite proud of the fact that this inference is "prior free", but the GVA strikes me as much a prior as any sort of smoothness or regularity assumption. This paper might have a more positive tone if this GVA is presented as a new, powerful prior, which is as effective as several of the priors of [3], and which could possibly be integrated into other algorithms such as [3].
- The GVA assumption, implemented here, seems heavily integrated into the optimization machinery. Is it possible for the assumption to be factored out and used in a more general framework, like [3]? If so, then it would be very easy to tease apart the success of this algorithm: does it work well because of the GVA, or because the optimization techniques deal with local minima well, or both? As the converse to this point, it would be interesting to see this ADMM inference approach using the priors of [3], to further probe the source of the results here.
Summary: This paper leverages a very smart and previously unexploited high-level vision insight, and uses very clever inference to make that insight practical.

Submitted by Assigned_Reviewer_9

The authors present a work in the context of estimating shape and illumination from a single shading image (an image of a constant albedo object).

The authors in [3] propose a computationally practical approach by adopting intricate assumptions over shape and illumination. The authors in [9] explore the simpler Generic Viewpoint Assumption (GVA) without proposing computationally practical approach. The contribution of the work under consideration is putting GVA to practical use and replacing thus intricate assumptions over shape and illumination with both conceptually and technically simpler GVA.

The authors contribute to a long standing and well studied basic low-level vision task. While approaching a rather specific problem, their contribution is both technical and conceptual, and may be significant to a subset of NIPS audience. The work is well presented.
Summary: The authors contribute to a long standing and well studied basic low-level vision task. While approaching a rather specific problem, their contribution is both technical and conceptual, and will be of interest to a part of NIPS audience.

Submitted by Assigned_Reviewer_14

This paper proposes a method for estimation of shape and illumination from shading by using the generic view assumption proposed in [9]. The authors formulate a cost function that consists of a data term and a regularization term based on the generic viewpoint assumption that depends on both shape and illumination. The cost is minimized with respect to these unknowns using a message passing variant for alternating direction method of multipliers. The experimental results on the MIT/Berkeley intrinsic images database show that the method performs close to the prior art method by Barron and Malik, but it does not outperform it.

The idea of using the generic viewpoint assumption (GVA) directly in the cost function for shape and illumination from shading is novel, to the best of my knowledge. Therefore, the idea presented in the paper is original. The addressed problem is also a very significant one in computer vision. However, the authors have not made their case to convince the reader why incorporating the GVA actually makes sense and why they expect it to outperform previous work. Particularly, they should explain why using GVA makes more sense than using natural shape statistics priors (e.g. shape smoothness). GVA states that the viewpoint is typically not in a special relation to the object, thus small variations in the viewpoint (or light source) will induce small variations in the object appearance. In that case, one can say that invariance to such small variations under GVA relies a lot on the natural statistics of the scene, such as smoothness. Hence, using GVA as a prior could indirectly assume shape smoothness as well. However, the smoothness then might be enforced only with respect to the viewpoint change considered in the GVA term in the cost function (in this paper only with respect to rotations). Could this interpretation explain the unusual streaks in the shape reconstruction results and the dependence of their orientation on the viewpoint? I think it is very important to discuss this issue in the paper, and justify why the GVA prior is better than the smoothness prior. Even though the authors state that their GVA is not a prior, looking at the cost function it seems to me that it is a prior.

The clarity of the paper should be significantly improved. Besides my comment related to the GVA assumption above, the authors need to put more effort into explaining the mathematical formulation of the GVA term in the cost function, given in the Eq.14. It is not clear at all how the authors arrive to this formulation from Eq.9-13. Moreover, how they obtain Eq.11-13 is not clear either. The supplemental material only provides the derivation of the linearization formula. There is also a typo in the supplemental in two equations that are not enumerated (please enumerate them!) and are between Eq.5 and Eq.6 - it should be h here instead of f. Overall, the mathematical derivation of the cost function in Eq.15 is not clear at all, in addition to the unclear motivation of using this cost.

The obtained results do not justify the method either. Numerical results show that the proposed method does not outperform state of the art, which makes one wonder why would someone use the proposed method. Visual comparisons are difficult to interpret since the proposed method gives shapes with strange artifacts. These are some streaks, whose orientation seems to be dependent on the viewpoint. These streaks are visible on the 3D shape renderings, but not in the rendered image from a specific viewpoint. This makes the reader wonder how would the rendered image look like from another viewpoint?

In detail:
Sec. 3. "We found it to be highly accurate for the models in our experiments" - what about the models not presented in the paper?
Sec. 4.1 - Authors need to be more careful with the language to make sure the reader understands that this was done in [9]. E.g. line 180: "We get the …" - Actually it has been obtained in [9]
In supple. mat. Eq.6 uses a variable 'x'. How is that obtained?
Sec.6 What is the mask and how it affects the results?
Sec.6. Why do you do a DC shift?
Summary: Overall, the paper proposes a novel idea of using the general viewpoint assumption in estimating shape and illumination from shading, but fails to justify this approach. The authors need to better motivate, explain and evaluate their method according to the comments above.
Author Feedback
Author rebuttal: We thank the reviewers for their detailed feedback.

R14 asked about the difference between the GVA term and a smoothness prior over the surface. GVA is not a surface smoothness prior. It is a term that depends on the formation model and the observed image. It is not a prior over the parameters we want to estimate, but, rather, a prior over a nuisance variable that we marginalize out (lighting or viewpoint). In that sense, we agree with R3 that the GVA can be seen as a form of a prior, and will edit the paper accordingly. In [9], Freeman has shown the existence of solution surfaces that are smooth but unstable (section 3.4), such solutions would thus would not be ruled out by the use of a surface smoothness prior alone.

R14 asked about the appearance of streaks in the shape reconstruction results. We note that the non-GVA results of Figures 3 and 4 show streaks, which are similar to artifacts shown in some of the results in [9] when a GVA term is not used. These streaks are significantly reduced in the GVA results in Figures 3 and 4, which is the point of using a GVA term. Remaining artifacts in the GVA results could be due to a combination of sub-optimal minima or overly sparse sampling of rotation angles. We also note that the streaks are a function of light direction (being parallel to the light isophotes), and not a function of viewpoint.

R14 asked why streaks visible in the 3D shape are not visible in the 2D rendered image: This is an important point: there are a large number of feasible explanations for the observed image which are completely wrong/unrealistic. The streaks are an example of such a solution; the light is precisely aligned with the streaks so they don't show in the rendered image. The GVA is *exactly* the term which tries to avoid such contrived solutions, and indeed, there are very few such artifacts when we use the GVA. Furthermore, the use of surface priors such as smoothness will not help with such solutions.

R14 has questions about the performance of the proposed method. We have compared with 32 different model/light combinations from two different datasets, and we outperform the SIRFS method of [3] on light estimation (Figure 2(c)) and depth estimation (Supplementary material). We chose not to add the priors in [3] into our cost term. While this might yield better results than either [3] or our method alone, we feel this would not shed light on the contribution of the GVA term itself, as performance can depend on tuning the relative balance of different terms.

R14 asked about the mask and how it affects the results. The mask is supplied with the data and provides the outline of the objects. We use this only to know which pixels to solve for in the shape estimation. Unlike [3], we did not use the contour normal information to help our shape estimation.